# Neural evidence of functional compensation for fluid intelligence in healthy ageing

Ethan Knights[1], Richard N Henson[1,2], Alexa Morcom[3], Daniel J Mitchell[1], Kamen A Tsvetanov[4,5]*

[1]Medical Research Council Cognition and Brain Sciences Unit, Cambridge, United Kingdom; [2]Department of Psychiatry, University of Cambridge, Cambridge, United Kingdom; [3]School of Psychology, University of Sussex, Brighton, United Kingdom; [4]Department of Clinical Neurosciences, University of Cambridge, Cambridge, United Kingdom; [5]Department of Psychology, University of Cambridge, Cambridge, United Kingdom

*For correspondence:
kat35@cam.ac.uk

Conflict of interest: The authors declare no competing financial interests.

Competing interest: The authors declare that no competing interests exist.

## eLife Assessment

This study provides an **important** advancement of knowledge by showing neural functional compensation in the brains of healthy older adults completing a fluid-intelligence task. Validated whole-brain voxel-wide analyses and multivariate Bayesian approaches provide **compelling** evidence that supports the claims of the authors. The work delivers methods for quantifying reserve and compensation in future studies and will be of interest to researchers in the field of the neuroscience of healthy aging.

**Abstract** Functional compensation is a common notion in the neuroscience of healthy ageing, whereby older adults are proposed to recruit additional brain activity to compensate for reduced cognitive function. However, whether this additional brain activity in older participants actually helps their cognitive performance remains debated. We examined brain activity and cognitive performance in a human lifespan sample ($N$ = 223) while they performed a problem-solving task (based on Cattell's test of fluid intelligence) during functional magnetic resonance imaging. Whole-brain univariate analysis revealed that activity in bilateral cuneal cortex for hard vs. easy problems increased both with age and with performance, even when adjusting for an estimate of age-related differences in cerebrovascular reactivity. Multivariate Bayesian decoding further demonstrated that age increased the likelihood that activation patterns in this cuneal region provided non-redundant information about the two task conditions, beyond that of the multiple demand network generally activated in this task. This constitutes some of the strongest evidence yet for functional compensation in healthy ageing, at least in this brain region during visual problem-solving.

## Introduction

Preventing cognitive decline in old age is a major public heath priority, which demands a better understanding of the neurophysiological changes that preserve cognitive function despite progressive brain atrophy (*Cabeza et al., 2018*; *Christensen et al., 2009*). Neuroimaging has facilitated the idea that the brain can flexibly respond to tissue loss (e.g., due to ageing) by recruiting additional brain activity to support cognitive functions (*Cabeza et al., 2018*; *Grady, 2012*). If this additional recruitment in

older adults improves their behavioural performance, it is argued that this reorganisation of brain function constitutes a functional compensation mechanism (*Cabeza, 2002*).

Fluid intelligence (i.e., solving novel abstract problems) is a cognitive function that shows one of the most consistent and largest decreases in older age (*Salthouse et al., 2008*; *Deary, 2012*; *Ghisletta et al., 2012*; *Kievit et al., 2014*; *Bors and Forrin, 1995*; *Salthouse and Pink, 2008*; *Schretlen et al., 2000*; *Clay et al., 2009*; *Kievit et al., 2018*). Functional (*Duncan and Owen, 2000*; *Gray et al., 2003*; *Lee et al., 2006*; *Crittenden et al., 2016*; *Tschentscher et al., 2017*) and structural (*Colom et al., 2009*; *Jauk et al., 2015*; *Chen et al., 2020*; *Paul et al., 2016*; *Zamroziewicz et al., 2018*) neuroimaging has shown that fluid intelligence tasks engage the multiple demand network (MDN; *Duncan, 2010*), which comprises lateral prefrontal, posterior parietal, and cingulate regions. MDN activation tends to decrease with age as measured, for example, with functional magnetic resonance imaging (fMRI) during problem-solving tasks that tax fluid intelligence such as the Cattell task (*Samu et al., 2017*; *Mitchell et al., 2023*). So far, these studies have examined age effects in core regions of the MDN but have not explicitly tested for functional compensation in other regions.

To search for brain regions that might support functional compensation, we conducted a whole-brain voxel-wise search for clusters that showed a positive relationship with both age and cognitive performance (i.e., classic univariate criteria for functional compensation; *Lövdén et al., 2010*; *Cabeza et al., 2018*). The dependent variable was the difference in fMRI activation for blocks of hard vs. easy odd-one-out problems (*Figure 1A*), as measured in 223 adults between 19 and 87 years of age, from Stage 3 of the Cambridge Centre for Ageing & Neuroscience (Cam-CAN) project (*Shafto et al., 2014*); performance was measured as the proportion of all problems correct. Second, we applied a multivariate Bayesian (MVB) approach (*Friston et al., 2008*) across all voxels within any candidate regions identified in the whole-brain search, to test whether multi-voxel patterns in these regions provided additional information about task difficulty, beyond that in the MDN. We predicted that, if a region were involved in functional compensation, the additional information it contains about the task would increase with age. To pre-empt the results, unlike in our previous applications of MVB (*Morcom and Henson, 2018*; *Knights et al., 2021*), we find one region – within the cuneus – that did show evidence of this additional multivariate information, supporting its role in functional compensation.

## Results

### Behavioural performance

As expected from prior studies, behavioural performance decreased with age during the fMRI scan on the modified version of the Cattell task (collapsed across hard and easy conditions; see Methods) (standardised coefficient = −5.65, $t(220)$ = −14, $p < 0.001$, $R^2$ = 0.48; *Figure 1B*, upper). There was a high correlation between performance measures from the fMRI version and standard version of the Cattell task when the same people performed the standard Cattell task outside the scanner 1–3 years previously ($r$ = 0.79, $p < 0.001$; *Figure 1B*, lower), suggesting that the version modified for fMRI was capturing the same cognitive ability.

### Univariate response

The [Hard > Easy] contrast showed bilateral activation across regions generally described as comprising the MDN (e.g., *Duncan, 2010*; *Smith et al., 2021*), including the inferior/middle frontal gyri, intra-parietal sulcus, anterior insula, and anterior cingulate cortex (*Figure 1C*). Additional activation was observed bilaterally in the inferior/ventral and lateral occipital temporal cortex (i.e., a cluster around the lateral occipital sulcus that extended anteriorly beyond the anterior occipital sulcus), likely due to the visual nature of the task.

To search for a potentially compensatory pattern of brain activation, we next overlaid maps that tested for positive effects of age (*Figure 2A*, green map) and performance (*Figure 2A*, red map) on the [Hard > Easy] contrast. While age and performance are negatively correlated (*Figure 1B*), their effects were estimated simultaneously via multiple regression, and so the activation maps reflect unique effects of each. As reported using related measures and overlapping samples of Cam-CAN participants (*Samu et al., 2017*; *Wu et al., 2023*; *Mitchell et al., 2023*), age-related increases in activity were widespread, including the precuneus, middle frontal gyrus, and supplementary motor area. Activity positively related to performance was found in many of the same regions that were

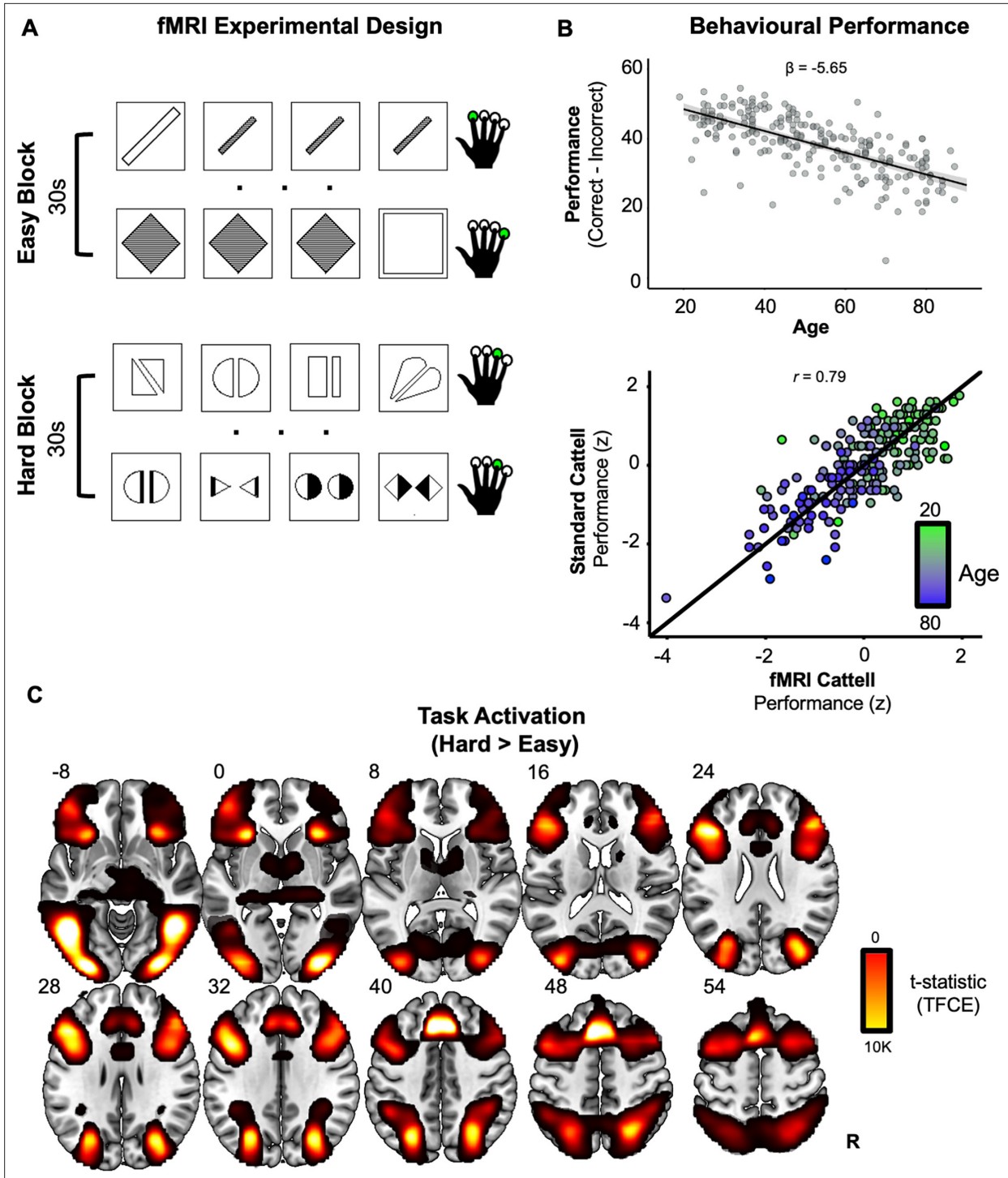

**Figure 1.** Experimental design, behavioural performance and task activation effects. (**A**) Functional magnetic resonance imaging (fMRI) version of Cattell task. On each trial (each row), participants select the odd-one-out from four panels with a single finger button-press (green circles). Condition blocks (30 s) alternate between easy vs. hard puzzles. (**B**) Behavioural age-related decline. Performance (correct minus incorrect in fMRI version of Cattell task) significantly declined linearly with age (upper). High reliability was observed between performance measures from the standard Cattell task and the modified version used for fMRI (lower). In the upper panel, the black line represents the fitted-regression estimates with shaded 95% confidence intervals. In the lower panel, the black line represents perfect correlation between the two Cattell versions. (**C**) Univariate task effect. Whole-brain voxel-wise activations for solving the puzzles in the hard, relative to easy, blocks, after threshold-free cluster enhanced (TFCE) correction (slices are labelled with z MNI coordinates).

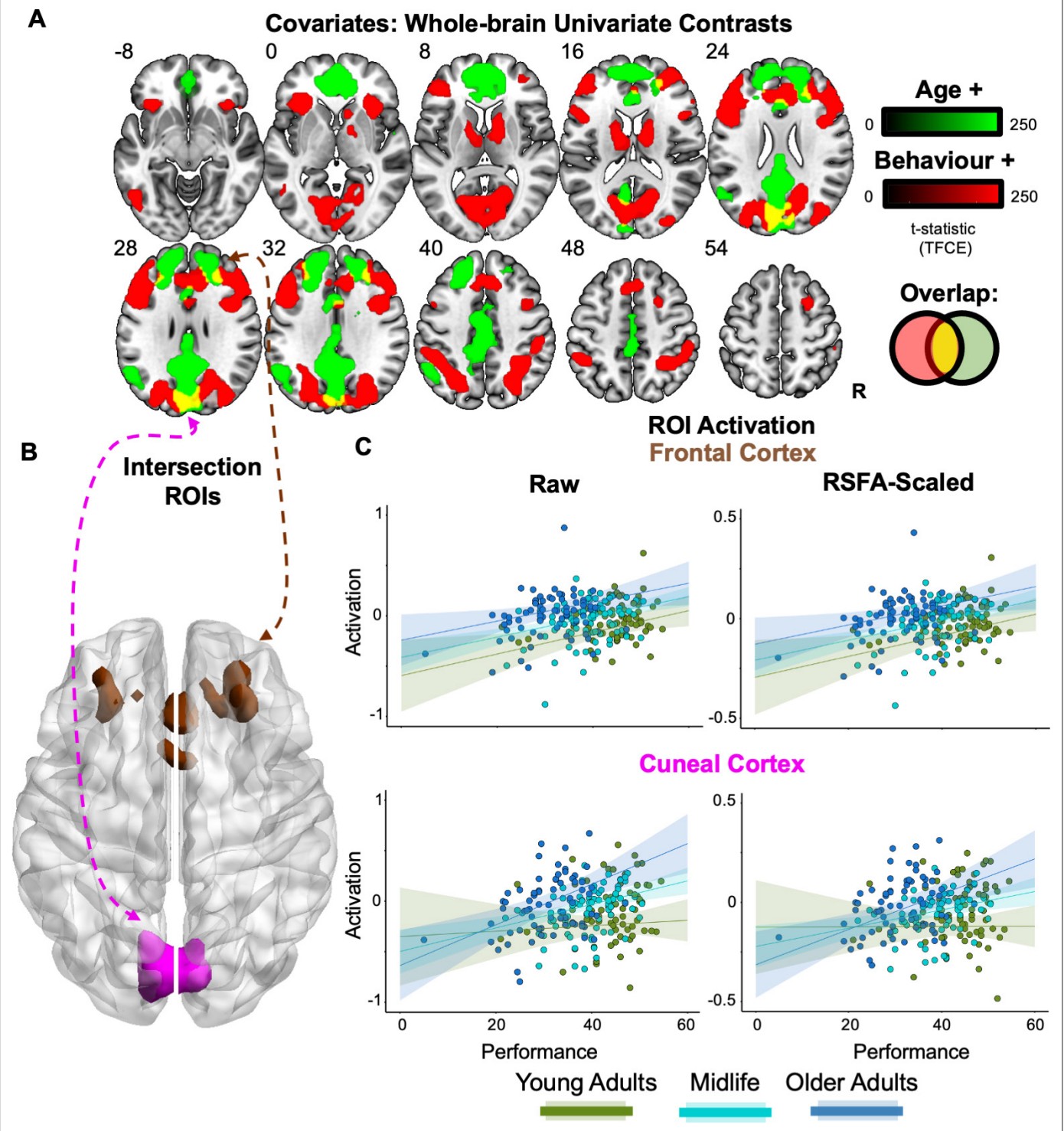

**Figure 2.** Univariate analysis. (**A**) Whole-brain effects of age and performance. Age (green) and performance (red) positively predicted unique aspects of increased task activation, with their spatial overlap (yellow) being overlaid on a template MNI brain, using p < 0.05 threshold-free cluster enhanced (TFCE) with z MNI coordinate labels. (**B**) Intersection regions of interest (ROIs). A bilateral cuneal (magenta) and frontal cortex (brown) ROI were defined from voxels that showed a positive and unique effect of both age and performance (yellow map in **A**). (**C**) ROI activation. Activation (raw = left; Resting State Fluctuation Amplitude [RSFA]-scaled = right) is plotted against behavioural performance based on a tertile split between three age groups (19–44, 45–63, and 64–87 years).

more active for hard vs. easy problems (i.e., inferior/middle frontal gyrus, anterior cingulate, superior parietal lobule; *Figure 1C*).

Crucially, two areas of the brain showed spatially overlapping positive effects of age and performance, which is suggestive of an age-related compensatory response (*Figure 2A*, yellow intersection). These were in bilateral cuneal cortex (*Figure 2B*, magenta) and bilateral frontal cortex (*Figure 2B*, brown), the latter incorporating parts of the middle frontal gyri and anterior cingulate. Therefore, based on traditional univariate analyses, these are two candidate regions for age-related functional compensation (*Cabeza and Dennis, 2013*; *Cabeza et al., 2018*). Accordingly, we defined regions of interest (ROIs) within these two regions using the overlap activation maps (see section: Regions of interest) to be used for subsequent uni- and multivariate analyses.

However, the two candidate compensation regions showed different patterns as a function of age and performance: whereas the frontal region showed additive effects of both variables (*Figure 2C*, upper), the cuneus region showed signs of an interaction (p = 0.028; though this would not survive correction for multiple comparisons across the two ROIs), whereby the relationship with performance was strongest in the oldest participants (and there was little sign of a performance relationship in the

**Table 1.** Standardised coefficients in multiple regression predicting functional magnetic resonance imaging (fMRI) activation (Hard − Easy) as a function of Age and Performance for the two regions of interest (ROIs) identified in *Figure 2*.

Note that the p-values for the main effects of Age and Performance are biased by the selection of these voxels. RSFA = scaled by Resting-State Fluctuation Amplitudes (see text).

| Region | Coefficient | Estimate | t value | p |
|---|---|---|---|---|
| Cuneal | | | | |
| | Constant term | −0.06 | −2.57 | 0.011 |
| | Age | 0.09 | 3.57 | <0.001 |
| | Performance | 0.08 | 3.21 | 0.002 |
| | Sex | −0.05 | −2.59 | 0.01 |
| | Age × Performance | 0.04 | 2.21 | 0.028 |
| Cuneal (RSFA) | | | | |
| | Constant term | −0.03 | −2.60 | 0.010 |
| | Age | 0.04 | 3.24 | 0.001 |
| | Performance | 0.04 | 3.11 | 0.002 |
| | Sex | −0.02 | −2.52 | 0.013 |
| | Age × Performance | 0.02 | 1.97 | 0.049 |
| Frontal | | | | |
| | Constant term | −0.03 | −2.02 | 0.045 |
| | Age | 0.08 | 4.24 | <0.001 |
| | Performance | 0.08 | 4.54 | <0.001 |
| | Sex | <0.001 | −0.35 | 0.728 |
| | Age × Performance | <0.001 | −0.15 | 0.882 |
| Frontal (RSFA) | | | | |
| | Constant term | −0.02 | −1.99 | 0.048 |
| | Age | 0.04 | 4.12 | <0.001 |
| | Performance | 0.04 | 4.48 | <0.001 |
| | Sex | 0.00 | −0.37 | 0.709 |
| | Age × Performance | 0.00 | −0.13 | 0.898 |

youngest participants; *Figure 2C*, lower). This is suggestive of compensatory activation only engaged by higher-performing older people in the cuneus specifically.

It has previously been shown that many effects of age on the blood oxygenation level-dependent(BOLD) signal measured by fMRI relate to vascular effects of ageing, rather than necessarily indicating differences in neural activity (*Tsvetanov et al., 2021a*). We therefore repeated the multiple regressions after scaling the Cattell activation effect by an estimate of the Resting State Fluctuation Amplitude (RSFA) for each ROI from an independent, resting-state scan for each participant. Previous work has shown that RSFA relates to age-related vascular differences (*Tsvetanov et al., 2021a*), but not neural differences (*Tsvetanov et al., 2015*; *Kumral et al., 2020*). Despite this RSFA adjustment, the pattern of effects remained similar in each ROI (*Table 1*; *Figure 2C*). This suggests that these effects of age (and the relationship with performance) are neural in origin. This check has not been performed in previous fMRI studies of age-related compensation, which could reflect vascular effects of ageing instead.

## MVB decoding

Next, we examined if these candidate compensation regions showed multivariate evidence of compensation. If their age- and performance-related activation reflects compensation, then multi-voxel analyses should show that this 'hyper-activation' carries additional information about the task,

**Table 2.** Standardised coefficients in multivariate Bayesian (MVB) multiple logistic regression analyses predicting boost likelihood as a function of age or the spread of voxel weights (with Sex and Mean Univariate Activation as covariates).

| Analysis | Model | Coefficient | Estimate | z/t-statistic | p |
|---|---|---|---|---|---|
| Boost likelihood | | | | | |
| | Cuneal ROI + Task-network (166 voxels each) | | | | |
| | | Constant term | 2.17 | 8.79 | <0.001 |
| | | Age | 0.79 | 3.23 | <0.001 |
| | | Sex | 0.08 | 0.37 | 0.714 |
| | | Univariate | −0.34 | −1.59 | 0.112 |
| | Frontal ROI + Task-network (85 voxels each) | | | | |
| | | Constant term | 2.20 | 9.30 | <0.001 |
| | | Age | 0.04 | 0.19 | 0.851 |
| | | Sex | 0.26 | 1.13 | 0.257 |
| | | Univariate | −0.54 | −2.33 | 0.020 |
| Spread (weights) | | | | | |
| | Cuneal ROI | | | | |
| | | Constant term | <−0.001 | <0.001 | >0.999 |
| | | Age | −0.06 | −0.84 | 0.403 |
| | | Sex | −0.05 | −0.64 | 0.521 |
| | | Univariate | 0.01 | 0.09 | 0.932 |
| | Frontal ROI | | | | |
| | | Constant term | <−0.001 | <0.001 | >0.999 |
| | | Age | −0.19 | −2.89 | 0.005 |
| | | Sex | −0.09 | −1.37 | 0.171 |
| | | Univariate | 0.09 | 1.33 | 0.185 |

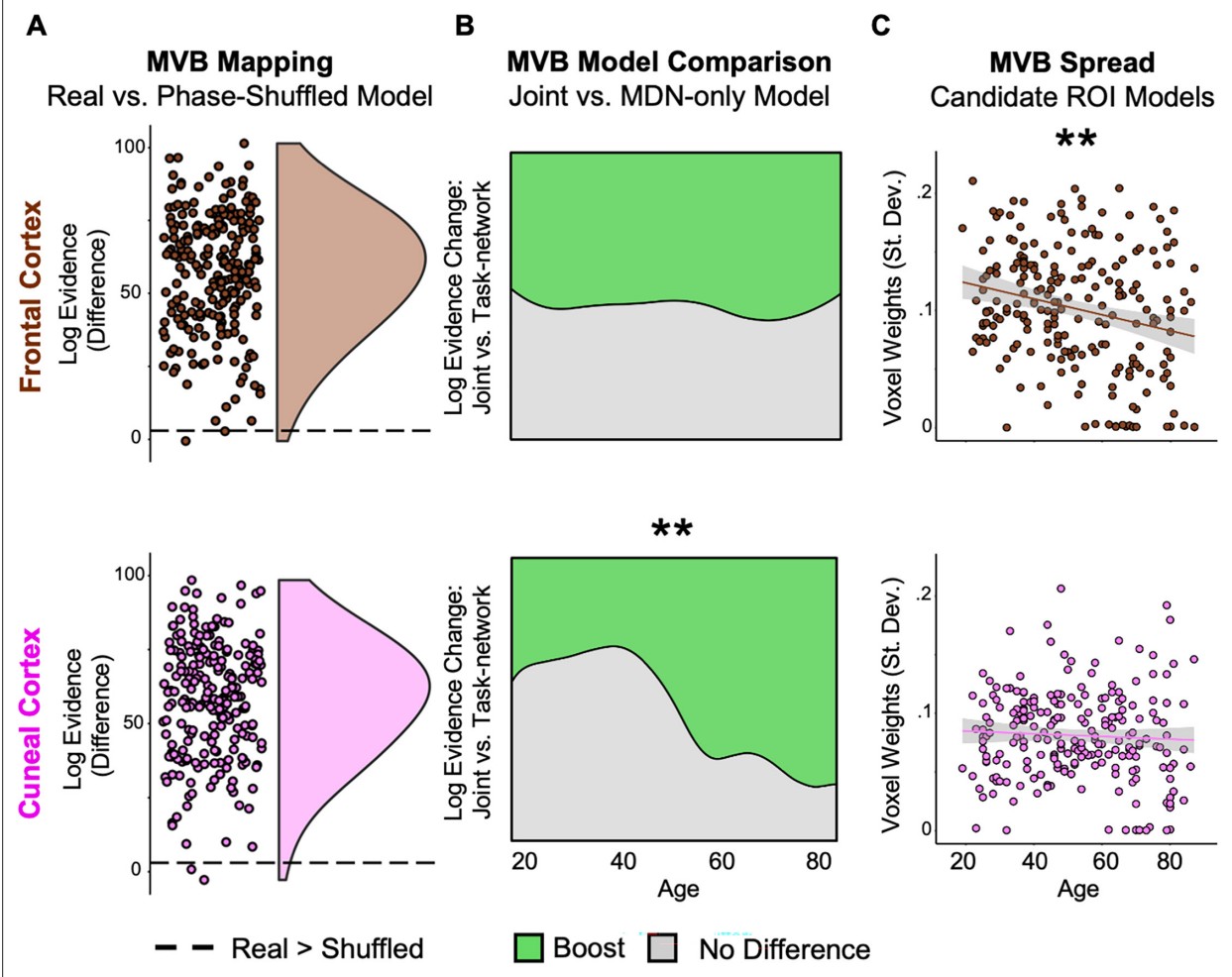

**Figure 3.** Multivariate analysis. (**A**) Multivariate Bayesian (MVB) decoding. Points represent the difference in log evidence per participant (for the real vs. shuffled model) for the joint model using activation patterns to decode the [Hard > Easy] contrast. (**B**) Boost likelihood model comparison. Across age, a smoothed density estimate represents the likelihood that there was a boost (of log evidence >3; green) or no difference (grey) to model evidence per participant when decoding models included activation patterns from either of the compensation regions of interest (ROIs) (**Figure 2B**) in addition to the multiple demand network (MDN) (**Figure 1C**), relative to a model that sampled only from the MDN. A significant positive linear effect of age on boost likelihood was observed for the cuneal (lower) but not frontal ROI (upper). (**C**) MVB spread. Points represent the spread (standard deviation) of multivariate responses, showing a reduction with age in frontal cortex. ** denotes p < 0.01.

over and above that already provided by the regions generally activated by the task (i.e., MDN). To test this, we applied MVB decoding of the [Hard >Easy] contrast.

We first implemented MVB with a 'joint model' that contained voxel activation patterns from (1) one of the potential compensation ROIs and (2) the same number of the most significant voxels in the MDN (defined by the orthogonal contrast of [Hard > Easy]; **Figure 1C**; see **Table 2** for voxel numbers). For each joint model (i.e., MDN voxels + cuneal or frontal voxels), we compared the log model evidence for the correct model to ones where the stimulus onsets were shuffled (i.e., to estimate a null distribution of model evidence). Across both joint models (MDN plus cuneal or frontal cortex), we found evidence of above-chance decoding (real vs. shuffled log-evidence difference >3; see Methods) for all except two participants. These participants (the two points below the y = 3 dashed line in **Figure 3A**, one of whom was the same across models) were removed (**Morcom and Henson, 2018**; **Knights et al., 2021**).

Having established that the task condition could be decoded from voxels in almost all participants, the critical test was whether age influenced the likelihood that adding voxel activation patterns from the 'compensatory' ROIs (i.e., joint model) would boost decoding accuracy relative to that for the MDN-only model. A positive age effect on boost likelihood would indicate that, the older someone

was, the more likely that activation patterns in the putative 'compensation ROI' would provide additional, non-redundant, task-relevant information, consistent with a compensatory role. In line with this compensation account, there was a significant positive effect of age (*Table 2*) on the likelihood that model performance was boosted (i.e., log-evidence change >3) by including voxel activation patterns from the cuneal ROI (*Figure 3B*, lower; odds ratio = 2.21). In other words, the amount of unique task information in the multi-voxel pattern within the cuneal ROI (above that present in the MDN) increased with age. In contrast, this analysis for the model containing the frontal ROI voxel activation patterns showed no effect of age (*Table 2*; *Figure 3B*, upper).

Note that, since this age effect in the cuneus was present even though the logistic regression model contained this ROI's univariate response as a covariate of no interest (*Table 2*), the effect of age on boost likelihood is unlikely to be due to differences in the overall signal-to-noise ratio (SNR) across ages.

As a final analysis, we also tested a more lenient definition of functional compensation, whereby the multivariate contribution from the 'compensation ROI' does not necessarily need to be above and beyond that of the task-relevant network (*Morcom and Henson, 2018*; *Knights et al., 2021*). To do this, we again assessed whether age was associated with an increase in the spread of the weights over voxels (see Methods), for smaller models containing only the cuneal or frontal ROI. This tested whether increased age led to more voxels carrying substantial information about task difficulty, a pattern predicted by functional compensation (but also consistent with non-specific additional recruitment). In this case, the results of this test did not support functional compensation, as there was no effect detected for the cuneal cortex and even a negative effect of age for the frontal cortex where the spread of the information across voxels was lower for older age (*Figure 3C*; *Table 2*). This said, it is worth highlighting that even if an ROI were to have shown an effect of age on this spread measure, it could instead be explained by a non-specific mechanism that recruits multiple regions in tandem (rather than reflecting compensation) as seen previously (*Knights et al., 2021*; also see *Morcom and Johnson, 2015*).

## Discussion

The existence of age-related functional compensation mechanisms remains a matter of debate in the cognitive neuroscience of healthy ageing. Here, we analysed fMRI data from a problem-solving (fluid intelligence) task and identified two brain regions (in bilateral cuneal and frontal cortex; *Figure 2A, B*) that satisfied traditional univariate criteria for functional compensation. After applying the multivariate criterion that a compensating region should possess additional information about the task, only the cuneal cortex showed an age-related increase in this additional information (*Figure 3B*), beyond that available in the generic task-activated regions (i.e., the MDN; *Figure 1C*). This is the first demonstration of increased multivariate information with age, since previous studies have shown evidence for no such multivariate increase associated with univariate age-related hyper-activation in other ROIs and tasks; leading to previous findings being interpreted in terms of neural inefficiency, rather than compensation (*Morcom and Henson, 2018*; *Knights et al., 2021*).

Why would the cuneal cortex demonstrate functional compensation when solving difficult visuospatial problems? Since the cuneus has a well-established role in visual attention (e.g., *Corbetta et al., 1998*), we hypothesise that the additional recruitment of this brain region facilitates concurrently attending to multiple features of the stimulus array, to correctly select the 'odd-one-out'. The recruitment of this brain region in older adults could drive changes in looking strategy (e.g., *Law et al., 1996*), where, for example, older adults compensate for their reduced visual short-term memory (*Mitchell et al., 2018*) – that is, difficulty sustaining representations of puzzle items – by using more or different saccades. This possibility is consistent with the greater cuneal activation that was observed for older adults who performed better at the task (*Figure 2C*). Future work pairing fMRI behavioural tasks with eye-monitoring could verify this proposed relationship between age, cuneus activation, overt attention, and fluid intelligence.

In line with this hypothesised role of the cuneal cortex, there is consistent functional (*Yin et al., 2015*; *Santarnecchi et al., 2017*) and structural (*Haier et al., 2004*; *Jauk et al., 2015*; *Chen et al., 2020*) neuroimaging evidence that link this brain region to aspects of fluid intelligence like rule application. Similarly, responses from sensory areas (like the secondary visual network that overlaps our cuneus ROI; *Ji et al., 2019*) have been shown to predict fluid intelligence performance (*Brumback*

*et al., 2004*). In ageing, it is well established that sensory and intellectual decline are correlated (see *Baltes and Lindenberger, 1997*), either because they share a common cause or because performance of fluid intelligence tasks is partially dependent on sensory processing (e.g., *Schneider and Pichora-Fuller, 2000*). While our data cannot tease apart these hypotheses, it may be that compensatory processes in the cuneal region reflect this shared age-related variance between sensory and higher-order cognitive tasks.

Though activation of the cuneal ROI increased with age, it is worth noting the constant term (reflecting the average across all ages) was negative (*Table 1*), suggesting that most people (other than the older ones) showed greater activation of this region for easy than hard problems. This is more difficult to reconcile with its activation reflecting visual attention or eye movements, since this would suggest greater visual attention/eye movements towards easy than hard problems in the young. One alternative possibility is active suppression of the cuneal region in the hard blocks, to avoid distraction (e.g., minimise attentional capture from neighbouring display panels while processing features in each panel). Thus, the age-related reduction in the Easy − Hard difference (leading to the positive correlation of the Hard − Easy difference with age) could reflect reduced ability to inhibit the cuneus during hard problems, consistent with the established age-related decline in the ability to suppress distracting information in complex stimuli (*Tsvetanov et al., 2013*; *Rey-Mermet and Gade, 2018*; *Bouhassoun et al., 2022*). However, it is not clear why this alternative account would predict a positive correlation between cuneal activity and task performance, given that greater suppression (in the Hard condition) would be expected to lead to better performance, but more negative activity values for the [Hard − Easy] contrast. Thus, we favour the explanation in terms of functional compensation.

Another possibility is that the age-related increases in fMRI activations (for hard vs. easy) in one or both of our ROIs do not reflect greater fMRI signal for hard problems in older than younger people, but rather lower fMRI signal for easy problems in the older. Without a third baseline condition, we cannot distinguish these two possibilities in our data. However, a reduced 'baseline' level of fMRI signal (e.g., for easy problems) in older people is consistent with other studies showing an age-related decline in baseline perfusion levels, coupled with preserved capacity of cerebrovascular reactivity to meet metabolic demands of neuronal activity at higher cognitive load (*Calautti et al., 2001*; *Jennings et al., 2005*). Though age-related decline in baseline perfusion occurs in the cuneal cortex (*Tsvetanov et al., 2021b*), the brain regions showing modulation of behaviourally relevant Cattell fMRI activity by perfusion levels did not include the cuneal cortex (*Wu et al., 2023*). This suggests that the compensatory effects in the cuneus are unlikely to be explained by age-related hypo-perfusion, consistent with the minimal effect here of adjusting for RSFA (*Figure 2C*). One final possibility is whether the observed boost in decoding from adding the cuneal ROI simply reflects less noisy task-related information (i.e., a better SNR) than the MDN and, consequently, the boosted decoding is the result of more resilient patterns of information (rather than the representation of additional information) based on a steeper age-related decline of SNR in the MDN. Overall then, as none of the explanations above agree with all aspects of the results, to functionally explain the role of the cuneal cortex in this task would require further investigation.

The age- and performance-related activation in our frontal region satisfied the traditional univariate criteria for functional compensation, but our MVB model comparison analysis showed that additional multivariate information beyond that in the MDN was absent in this region, which is inconsistent with the strongest definition of compensation. In fact, the results from the spread analysis showed that as age increased, this frontal area processed less, rather than more, multivariate information about the cognitive outcome (*Figure 3C*) as previously observed in two (memory) tasks for a comparable ROI within the same Cam-CAN cohort (*Morcom and Henson, 2018*).

This pattern of results suggests that traditional univariate criteria alone are not sufficient for identifying functional compensation. Similar univariate effects have been found in previous studies (though with smaller samples), where lateral and medial frontal areas show increased activation during healthy ageing across a range of tasks, including those related to executive control or attention (e.g., *Spreng et al., 2010*; for a review, see *Spreng et al., 2010*; also see *Raz et al., 2008*, for a neuroanatomical link). Patients with brain damage also demonstrate increased frontal activation during language and semantic processing (*Brownsett et al., 2014*; *Rice et al., 2018*) indicating that this mechanism might be a response to brain atrophy generally. Instead, our results suggest that this frontal hyper-activation in older adults reflects 'inefficient' processing, in terms of more neural resources being needed to

perform the task (i.e., for hard vs. easy problems). In fact, neural inefficiency was our favoured interpretation of previous cases when MVB showed no age-related boost, in frontal (*Morcom and Henson, 2018*) or motor (*Knights et al., 2021*) regions. From these studies, and all previous fMRI or positron emission tomography (PET) studies that showed age-related hyper-activation, it was not known whether the increased activations reflected greater neural inefficiency, or greater haemodynamic resources needed for the same level of neural activity (i.e., vascular rather than neural inefficiency). Here, we showed for the first time that the age-related increase in both ROIs remained even after adjusting for RSFA (*Table 1*), suggesting that this hyper-activation reflects neural rather than vascular inefficiency.

This said, univariate criteria for functional compensation will continue to play a role in hypothesis testing. For instance, the over-additive interaction observed in the cuneal cortex – where the increase in activity with better performance is more pronounced in older adults – offers evidence of compensation compared to the simple additive effect of age and performance observed in the frontal cortex (*Figure 2C*). However, the conclusions that can be drawn from age-related differences in cross-sectional associations of brain and behaviour are limited, mainly because individual performance differences are largely lifespan-stable (see *Lindenberger et al., 2011*; *Morcom and Johnson, 2015*). So far, the two studies that have combined these univariate, behavioural, and multivariate approaches to assess functional compensation (i.e., *Knights et al., 2021*; the present study) have generally found converging evidence regardless of the method used. However, it is important to note that the MVB approach uniquely shifts the focus from individual differences to the specific task-related information that compensatory neural activations are assumed to carry and provides a specific test of region- (or network-) unique information. With further studies, it may also be that multivariate approaches prove more sensitive for detecting compensation effects than when using mean responses over voxels (e.g., *Friston et al., 1995*) particularly since over-additive effects are challenging to observe because compensatory effects are typically 'partial' and do not fully restore function (for review see *Scheller et al., 2014*; *Morcom and Johnson, 2015*). Within the multivariate analysis options themselves, it is also interesting to highlight that the stringent MVB boost likelihood analysis could detect functional compensation unlike the more lenient analysis focusing on the spread of MVB voxel weights. This suggests the importance of including task-relevant network responses when building decoding models to assess compensation.

In *Morcom and Henson, 2018*, we did not explicitly test for a relationship between activation and (memory) performance, and in *Knights et al., 2021*, we failed to find any relationship between (ipsilateral motor) activation and various (motor) performance measures. In the present study, it may be that the age-related frontal hyper-activation is caused by neural inefficiency, yet the degree of overall activation still relates to (lifespan-stable) problem-solving performance. Converging with the lack of additional multivariate information, this suggests that the frontal region does not show a compensatory response.

In summary, we propose that our results in the cuneus represent the most compelling evidence to date for functional compensation in healthy ageing, with further work needed to determine the precise function of this region in problem-solving tasks like that examined here. Together with the results in prefrontal cortex, the data also suggest that specific compensatory neural responses can coexist with inefficient neural function in older people.

## Methods
### Participants
A healthy population-derived adult lifespan human sample ($N$ = 223; ages approximately uniformly distributed from 19 to 87 years; females = 112; 50.2%) was collected as part of the Cam-CAN study (Stage 3 cohort; *Shafto et al., 2014*). Participants were fluent English speakers in good physical and mental health, based on the Cam-CAN cohort's exclusion criteria which includes poor mini mental state examination, ineligibility for MRI and medical, psychiatric, hearing, or visual problems. Throughout analyses, age is defined at the Home Interview (Stage 1; *Shafto et al., 2014*). The study was approved by the Cambridgeshire 2 (now East of England – Cambridge Central) Research Ethics Committee (reference: 10/H0308/50) and participants provided informed written consent. Further demographic information of the sample is reported in *Wu et al., 2023* and is openly available (see

section: Data availability) with a recent report indicating the representativeness of the sample across sexes (*Green et al., 2018*).

## Materials and procedure

A modified version of the odd-one-out subtest of the standardised Cattell Culture Fair Intelligence test (Scale 2; *Cattell, 1971*; *Cattell and Cattell, 1973*) was developed for use in the scanner (*Woolgar et al., 2013*; *Samu et al., 2017*; *Wu et al., 2023*). Participants were scanned while performing the problem-solving task where, on each trial, four display panels were presented in a horizontal row (*Figure 1A*) in the centre of a screen that was viewed through a head-coil mounted mirror. Participants were required to make a button-press response to identify the mismatching panel that was unique in some way from the other three (based on either a figural, spatial, complex, or abstract property).

In a block design, participants completed eight 30-s blocks which contained a series of puzzles from one of two difficulty levels (i.e., four hard and four easy blocks completed in an alternating block order; *Figure 1A*). The fixed block time allowed participants to attempt as many trials as possible. Therefore, to balance speed and accuracy, behavioural performance was measured by subtracting the number of incorrect from correct trials and averaging over the hard and easy blocks independently (i.e., ((hard correct − hard incorrect) + (easy correct − easy incorrect))/2; *Samu et al., 2017*). For assessing reliability and validity, behavioural performance (total number of puzzles correct) was also collected from the same participants during a full version of the Cattell task (Scale 2 Form A) administered outside the scanner at Stage 2 of the Cam-CAN study (*Shafto et al., 2014*). Both the in- and out-of-scanner measures were $z$-scored. We excluded participants ($N = 28$; 17 females) who performed at chance level ((correct + incorrect)/incorrect <0.5) on the fMRI task, leading to the same subset as reported in *Samu et al., 2017*.

## Data acquisition and pre-processing

The MRI data were collected using a Siemens 3T TIM TRIO system with a 32-channel head-coil. A T2*-weighted echoplanar imaging sequence was used to collect 150 volumes, each containing 32 axial slices (acquired in descending order) with slice thickness of 3.0 mm and an interslice gap of 25% for whole-brain coverage (repetition time, RT = 2000 ms; echo time, TE = 30 ms; flip angle = 78°; field of view, FOV = 192 mm × 192 mm; voxel-size 3 × 3 × 3.75 mm). Higher resolution (1 mm × 1 mm × 1 mm) T1- and T2-weighted structural images were also acquired (to aid registration across participants).

MR data pre-processing and univariate analysis were performed with SPM12 software (Wellcome Department of Imaging Neuroscience, London, https://www.fil.ion.ucl.ac.uk/spm/), release 4537, implemented in the AA 4.0 pipeline (*Cusack et al., 2014*) described in *Taylor et al., 2017*. Specifically, structural images were rigid-body registered to an MNI template brain, bias corrected, segmented, and warped to match a grey matter template created from the whole Cam-CAN Stage 2 sample using DARTEL (*Ashburner, 2007*; *Taylor et al., 2017*). This template was subsequently affine transformed to standard Montreal Neurological Institute (MNI) space. The functional images were spatially realigned, interpolated in time to correct for the different slice acquisition times, rigid-body coregistered to the structural image, transformed to MNI space using the warps and affine transforms from the structural image, and resliced to 3 mm × 3 mm × 3 mm voxels.

## Univariate analysis

For participant-level modelling, a regressor for each condition was created by convolving boxcar functions of 30-s duration for each block with SPM's canonical haemodynamic response function, using a general linear model (GLM). Additional regressors were included in each GLM to capture residual movement-related artifacts, including six representing the *x/y/z* rigid-body translations and rotations (estimated in the realignment stage). Finally, the data were scaled to a grand mean of 100 over all voxels and scans within a session, and the model was fit to the data in each voxel. The autocorrelation of the error was estimated using an AR(1)-plus-white-noise model, together with a set of cosines that functioned to high-pass filter the model and data to 1/128 Hz, that were estimated using restricted maximum likelihood. The estimated error autocorrelation was then used to 'prewhiten' the model and data, and ordinary least squares used to estimate the model parameters. The contrast of parameter estimates for the hard and easy conditions, per voxel and participant, was then calculated

and combined in a group GLM with independent regressors for age and in-scanner behavioural performance.

## Univaraite region of interest (ROI) analysis

All ROIs were defined by selecting activated voxels from a group-level GLM (see *Table 2* for number of voxels within ROIs). The two ROIs that were tested as candidate regions for functional compensation (i.e., cuneal cortex and frontal cortex) were defined by contiguous voxels that were significantly positively related to the independent effects of both age and performance (see *Figure 2*). The MDN was defined by the selecting suprathreshold voxels activated by the [Hard vs. Easy] contrast from the Cattell task. For MVB analysis (see below), a subset of the highest activated voxels within the MDN were taken to match the number of voxels with that of the 'compensation ROI' being tested (see *Figure 3*; *Table 2*).

For the ROI-based multiple regressions, the activation was averaged across voxels (i.e., mean difference in parameter estimates for Hard – Easy conditions) for each ROI and participant (*Figure 2*, *Table 2*). In the case of RSFA-scaled multiple regression, we used RSFA calculated from independent resting-state scans (see *Tsvetanov et al., 2015*) to scale the task-related BOLD response (by dividing the Hard – Easy difference in parameter estimates for each voxel by the RSFA value at the same voxel).

## Multivariate Bayesian (MVB) analysis

A series of MVB models were fit to assess the information about task condition that was represented in each ROI or combination of ROIs. Each MVB decoding model is based on the same design matrix of experimental variables used in the univariate GLM, but the mapping is reversed; many physiological data features (fMRI activity in multiple voxels) are used to predict a psychological target variable (*Friston et al., 2008*). This target (outcome) variable is specified as the contrast [Hard > Easy] with all covariates removed from the predictor variables.

Each MVB model was fit using a parametric empirical Bayes approach, in which empirical priors on the data features (voxel-wise activity) are specified in terms of spatial patterns over voxel features and the variances of the pattern weights. As in earlier work (*Morcom and Henson, 2018*; *Knights et al., 2021*), we used a sparse spatial prior in which 'patterns' are individual voxels. Since these decoding models are normally ill-posed (with more voxels than scans), these spatial priors on the patterns of voxel weights regularise the solution.

The pattern weights specifying the mapping of data features to the target variable are optimised with a greedy search algorithm using a standard variational scheme (*Friston et al., 2008*) which was particularly appropriate given the large dataset. This is achieved by maximising the free energy, which provides an upper bound on the log of the Bayesian model evidence (the marginal probability of the data given that model). The evidence for different models predicting the same psychological variable can then be compared by computing the difference in log evidences, which is equivalent to the log of the Bayes factor (*Friston et al., 2008*; *Chadwick et al., 2012*; *Morcom and Friston, 2012*).

The outcome measure was the log evidence for each model (*Morcom and Henson, 2018*; *Knights et al., 2021*). To test whether activity from an ROI is compensatory, we used an ordinal boost measure (*Morcom and Henson, 2018*; *Knights et al., 2021*) to assess the contribution of that ROI for the decoding of task-relevant information (*Figure 3B*). Specifically, Bayesian model comparison assessed whether a model that contains activity patterns from a compensatory ROI and the MDN (i.e., a joint model) boosted the prediction of task-relevant information relative to a model containing the MDN only. The compensatory hypothesis predicts that the likelihood of a boost to model decoding will increase with older age. The dependent measure, for each participant, was a categorical recoding of the relative model evidence to indicate the outcome of the model comparison. The three possible outcomes were: a boost to model evidence for the joint vs. MDN-only model (difference in log evidence >3), ambiguous evidence for the two models (difference in log evidence between −3 and 3), or a reduction in evidence for the joint vs. MDN-only model (difference in log evidence <−3). These values were selected because a log difference of three corresponds to a Bayes factor of 20, which is generally considered strong evidence (*Lee and Wagenmakers, 2014*). Furthermore, with uniform priors, this chosen criterion corresponds to a p-value of <~0.05 (since the natural logarithm of 20 equals three, as evidence for the alternative hypothesis). A reduction in model evidence was not observed in the current study.

For this MVB boost analysis, participants were only included if their data allowed reliable decoding by the joint model (*Morcom and Henson, 2018*; *Knights et al., 2021*). To determine this, we contrasted the log evidence for the joint model with that from models in which the design matrix (and therefore the target variable) was randomly phase shuffled 20 times. The definition of reliable was based on a mean of 3 or more in the difference of log evidence between the true and shuffled model (*Morcom and Henson, 2018*; *Figure 3A*). Note that decoding is performed after removing the mean across voxels (i.e., MVB results are independent of the results in the univariate analyses presented in *Figure 1C* and *Table 1*).

Alongside the MVB boost analysis, we also included an additional measure using the spread (standard deviation) of voxel classification weights (*Morcom and Henson, 2018*). This measure indexes the absolute amplitude of voxel contributions to the task, reflecting the degree to which multiple voxels carry substantial task-related information. When related to age this can serve as a multivariate index of information distribution, unlike univariate analyses. However, it is worth highlighting that even if an ROI shows an effect of age on this spread measure, such an effect could instead be explained by a non-specific mechanism that represents the same information in tandem across multiple regions (rather than reflecting compensation) as seen previously (*Knights et al., 2021*; also see *Morcom and Johnson, 2015*). Thus, it is the MVB boost analysis that is the most compelling assessment of functional compensation because it can directly detect novel information representation.

### Experimental design and statistical analysis

Continuous age and behavioural performance variables were standardised and treated as linear predictors in multiple regression throughout the behavioural (*Figure 1B*), wholebrain voxel-wise (*Figures 1C and 2A*), univariate (*Table 1*; *Figures 1B and 2B*), and MVB (*Table 2*; *Figure 3*) analyses. Throughout, sex was included as a covariate. The models, including interaction terms, can be described, according to *Wilkinson and Rogers, 1973* notation, as *activity ~ age * performance + covariates* (which is equivalent to *activity ~ age:performance + age + performance + covariates*), allowing us to examine the unique variance explained by each predictor (*Table 1*) and to control for their shared variance. For whole-brain voxel-wise analyses, clusters were estimated using threshold-free cluster enhancement (*Smith and Nichols, 2009*) with 2000 permutations and the resulting images were thresholded at a *t*-statistic of 1.97 before interpretation. Bonferroni correction was applied to a standard alpha = 0.05 based on the two ROIs (cuneal and frontal) that were examined. For Bayes factors, interpretation criteria norms were drawn from *Jarosz and Wiley, 2014*.

### Acknowledgements

For the purpose of open access, the author has applied a Creative Commons Attribution (CC BY) licence to any Author Accepted Manuscript version arising from this submission. The Cambridge Centre for Ageing and Neuroscience (Cam-CAN) research was supported by the Biotechnology and Biological Sciences Research Council (Grant No. BB/H008217/1). The project has also received funding from the European Union's Horizon 2020 research and innovation programme ('LifeBrain', Grant Agreement No. 732592), which supported EK; KAT was supported by the Guarantors of Brain (G101149) and Alzheimer's Society (Grant No. 602). Corporate Cam-CAN authorship membership includes: Project principal personnel: Lorraine K Tyler, Carol Brayne, Edward T Bullmore, Andrew C Calder, Rhodri Cusack, Tim Dalgleish, John Duncan, Richard N Henson, Fiona E Matthews, William D Marslen-Wilson, James B Rowe, Meredith A Shafto; Research Associates: Karen Campbell, Teresa Cheung, Simon Davis, Linda Geerligs, Rogier Kievit, Anna McCarrey, Abdur Mustafa, Darren Price, David Samu, Jason R Taylor, Matthias Treder, Kamen A Tsvetanov, Janna van Belle, Nitin Williams, Daniel Mitchell, Ethan Knights; Research Assistants: Lauren Bates, Tina Emery, Sharon Erzinçlioglu, Andrew Gadie, Sofia Gerbase, Stanimira Georgieva, Claire Hanley, Beth Parkin, David Troy; Affiliated Personnel: Tibor Auer, Marta Correia, Lu Gao, Emma Green, Rafael Henriques; Research Interviewers: Jodie Allen, Gillian Amery, Liana Amunts, Anne Barcroft, Amanda Castle, Cheryl Dias, Jonathan Dowrick, Melissa Fair, Hayley Fisher, Anna Goulding, Adarsh Grewal, Geoff Hale, Andrew Hilton, Frances Johnson, Patricia Johnston, Thea Kavanagh-Williamson, Magdalena Kwasniewska, Alison McMinn, Kim Norman, Jessica Penrose, Fiona Roby, Diane Rowland, John Sargeant, Maggie Squire, Beth Stevens, Aldabra Stoddart, Cheryl Stone, Tracy Thompson, Ozlem Yazlik; and administrative staff: Dan Barnes, Marie Dixon, Jaya Hillman, Joanne Mitchell, Laura Villis.

## Additional information

### Funding

| Funder | Grant reference number | Author |
| --- | --- | --- |
| Biotechnology and Biological Sciences Research Council | BB/H008217/1 | Richard N Henson |
| Horizon 2020 Framework Programme | 732592 | Richard N Henson |
| Alzheimer's Society | 602 | Kamen A Tsvetanov |
| Guarantors of Brain | G101149 | Kamen A Tsvetanov |

The funders had no role in study design, data collection, and interpretation, or the decision to submit the work for publication.

### Author contributions

Ethan Knights, Conceptualization, Resources, Data curation, Software, Formal analysis, Validation, Investigation, Visualization, Methodology, Writing – original draft, Project administration, Writing – review and editing; Richard N Henson, Conceptualization, Resources, Software, Supervision, Funding acquisition, Investigation, Methodology, Writing – original draft, Project administration, Writing – review and editing; Alexa Morcom, Conceptualization, Software, Methodology, Writing – original draft, Writing – review and editing; Daniel J Mitchell, Conceptualization, Methodology, Writing – original draft, Writing – review and editing; Kamen A Tsvetanov, Conceptualization, Data curation, Supervision, Funding acquisition, Investigation, Methodology, Writing – original draft, Project administration, Writing – review and editing

### Author ORCIDs

Ethan Knights ⓘ https://orcid.org/0000-0001-6078-9160
Daniel J Mitchell ⓘ https://orcid.org/0000-0001-8729-3886
Kamen A Tsvetanov ⓘ https://orcid.org/0000-0002-3178-6363

### Ethics

The study was approved by the Cambridgeshire 2 (now East of England – Cambridge Central) Research Ethics Committee and participants provided informed written consent (reference: 10/H0308/50).

Reviewer #1 (Public Review): https://doi.org/10.7554/eLife.93327.3.sa1
Reviewer #2 (Public Review): https://doi.org/10.7554/eLife.93327.3.sa2
Reviewer #3 (Public Review): https://doi.org/10.7554/eLife.93327.3.sa3
Author response https://doi.org/10.7554/eLife.93327.3.sa4

## Additional files

### Supplementary files
MDAR checklist

### Data availability

Raw and minimally pre-processed MRI (i.e., from automatic analysis; *Taylor et al., 2017*) and behavioural data are available from https://camcan-archive.mrc-cbu.cam.ac.uk/dataaccess/. The raw data is publicly available for the purpose of scientific investigation or the planning of clinical research studies, subject to a data usage agreement. The univariate and multivariate ROI data, and behavioural data, can be downloaded from the Open Science Framework, which includes Cam-CAN participant identifiers allowing the retrieval of any additional demographic data (https://osf.io/v7kmh), while the analysis code is available on GitHub (https://github.com/ethanknights/Knightsetal_fMRI-Cattell-Compensation; copy archived at *Knights, 2024*).

The following dataset was generated:

| Author(s) | Year | Dataset title | Dataset URL | Database and Identifier |
|---|---|---|---|---|
| Knights E, Henson R, Morcom A, Tsvetanov K | 2022 | Neural evidence of functional compensation for fluid intelligence in healthy ageing | https://osf.io/v7kmh/ | Open Science Framework, v7kmh |

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
