## [Editor Report · eLife Assessment]

This study provides an **important** advancement of knowledge by showing neural functional compensation in the brains of healthy older adults completing a fluid-intelligence task. Validated whole-brain voxel-wide analyses and multivariate Bayesian approaches provide **compelling** evidence that supports the claims of the authors. The work delivers methods for quantifying reserve and compensation in future studies and will be of interest to researchers in the field of the neuroscience of healthy aging.

---

## [Referee Report · Reviewer #1 (Public review)]

This work addresses how to quantify functional compensation throughout the aging process and identifies brain regions that engage in compensatory mechanisms during the Cattell task, a measure of fluid cognition. The authors find that regions of the frontal cortex and cuneus showed unique effects of both age and performance. Interestingly, these two regions demonstrated differential activation patterns taking into account both age and performance. Specifically, the researchers found that the relationship between performance and activation in the cuneal ROI was strongest in older adults, however, this was not found in younger adults. These findings suggest that specifically within the cuneus, greater activation is needed by older adults to maintain performance, suggestive of functional compensation.

The conclusions derived from the study are well supported by the data. The authors validated the use of the in-scanner Cattell task by demonstrating high reliability in the same sample with the standard out-of-scanner version. Some strengths of the study include the large sample size and wide age range of participants. The authors use a stringent Bayes factor of 20 to assess the strength of evidence. The authors used a whole-brain approach to define regions of interest (ROIs) based on activation patterns that were jointly related to age and performance. Overall, the methods are technically sound and support the authors' conclusions.

Comment from Reviewing Editor: In the revised manuscript, the authors have addressed the weaknesses previously identified by reviewer 1.

---

## [Referee Report · Reviewer #2 (Public review)]

This work by Knights et al., makes use of the Cam-CAN dataset to investigate functional compensation during a fluid processing task in older adults, in a fairly large sample of approximately 200 healthy adults ranging from 19 to 87. Using univariate methods, the authors identify two brain regions in which activity increases as a function of both age and performance and conduct further investigations to assess whether the activity of these regions provides information regarding task difficulty. The authors conclude that the cuneal cortex - a region of the brain previously implicated in visual attention - shows evidence of compensation in older adults.

The conclusions of the paper are well supported by the data, and the authors use appropriate statistical analyses. The use of multivariate methods over the last 20 years has demonstrated many effects that would have been missed using more traditional univariate analysis techniques. The data set is also of an appropriate size, and as the authors note, fluid processing is an extremely important domain in the field of cognition in aging, due to its steep decline over aging.

Comment from Reviewing Editor: It would have been nice to see an analysis of a more crystallised intelligence task included too, as a contrast since this is an area that does not demonstrate such a decline (and perhaps continues to improve over aging). This comment does not take away the important contributions of the manuscript.

---

## [Referee Report · Reviewer #3 (Public review)]

This neuroimaging study investigated how brain activity related to visual pattern-based reasoning changes over the adult lifespan, addressing the topic of functional compensation in older age. To this end, the authors employed a version of the Cattell task, which probes visual pattern recognition for identifying commonalities and differences within sets of abstract objects in order to infer the odd object among a given set. Using a state-of-the-art univariate analysis approach on fMRI data from a large lifespan sample, the authors identified brain regions in which the activation contrast between hard and easy Cattell task conditions was modulated by both age and performance. Regions identified comprised prefrontal areas and bilateral cuneus. Applying a multivariate decoding approach to activity in these regions, the authors went on to show that only in older adults, the cuneus, but not the prefrontal regions, carried information about the task condition (hard vs. easy) beyond that already provided by activity patterns of voxels that showed a univariate main effect of task difficulty. This was taken as compelling evidence for task-specific compensatory activity in the cuneus in advanced age.

The study is well-motivated and well-written. The authors used appropriate, rigorous methods that allowed them to control for a range of possible confounds or alternative explanations. Laudable aspects include the large sample with a wide and even age distribution, the validation of the in-scanner task performance against previous results obtained with a more standard version outside the scanner, and the control for vascular age-related differences in hemodynamic activity via a BOLD signal amplitude measure obtained from a separate resting-state fMRI scan. Overall, the conclusions are well-supported by the data.

Comment from Reviewing Editor: The revised manuscript has addressed the points raised during the review of the original submission.

---

## [Author Response]

The following is the authors’ response to the original reviews.

**Reviewer #1:**
Point 1: While the manuscript is methodologically sound, the following aspects of image acquisition and data analysis need to be clarified to ensure replicability and reproducibility. The authors state that the sample is a "population-derived adult lifespan sample", the lack of demographic information makes it impossible to know if the sample is truly representative. Though this may seem inconsequential, education may impact both cognitive performance and functional activation patterns. Moreover, the authors do not report race/ethnicity in the manuscript. This information is essential to ensure representativeness in the sample. It is imperative that barriers to study participation within minoritized groups are addressed to ensure rigor and reproducibility of findings.

First, the section *Methods*-*Participants* has been updated to refer readers to a prior article where the sample’s demographics are broken down into nine decile age groups (see Wu et al. 2023 Table 1), including information about their education levels. Secondly, we have updated the Data Availability section text to indicate that all Cam-CAN IDs are included in the available OSF datasets, allowing anyone to verify additional participant demographics described in the Cam-CAN protocol article (Shafto et al., 2014). Third, we have updated the Participants section text to refer to another prior study that reported on the representativeness of the Cam-CAN sample indicating that at least some elements of the sample have been independently deemed as representative (e.g., Sex).

Page-24

“A healthy population-derived adult lifespan human sample (N = 223; ages approximately uniformly distributed from 19 - 87 years; females = 112; 50.2%) was collected as part of the Cam-CAN study (Stage 3 cohort; Shafto et al., 2014). Participants were fluent English speakers in good physical and mental health, based on the Cam-CAN cohort’s exclusion criteria which includes poor mini mental state examination, ineligibility for MRI and medical, psychiatric, hearing or visual problems. Throughout analyses, age is defined at the Home Interview (Stage 1; Shafto et al., 2014). The study was approved by the Cambridgeshire 2 (now East of England–Cambridge Central) Research Ethics Committee and participants provided informed written consent. Further demographic information of the sample is reported in Wu et al. (2023) and is openly available (see section Data Availability) with a recent report indicating the representativeness of the sample across sexes (Green et al., 2018).”

Page-30

“Raw and minimally pre-processed MRI (i.e., from automatic analysis; Taylor et al., 2017) and behavioural data are available by submitting a data request to Cam-CAN (https://camcan-archive.mrc-cbu.cam.ac.uk/dataaccess/). The univariate and multivariate ROI data, and behavioural data, can be downloaded from the Open Science Framework, which includes Cam-CAN participant identifiers allowing the retrieval of any additional demographic data (https://osf.io/v7kmh), while the analysis code is available on GitHub.”

Point 2: For the whole-brain analysis in which the ROIs were derived, the authors used a threshold-free cluster enhancement (TFCE; Smith & Nichols 2009). The methodological paper cited suggests that individuals' TCFE image should still be corrected for multiple comparisons using the following: "to correct for multiple comparisons, one [...] has to build up the null distribution (across permutations of the input data) of the maximum (across voxels) TFCE score, and then test the actual TFCE image against that. Once the 95th percentile in the null distribution is found then the TFCE image is simply thresholded at this level to give inference at the p < 0.05 (corrected) level." (Smith & Nichols, 2009). Although the authors mention that clusters were estimated using 2000 permutations, there is no mention of the TFCE image itself being thresholded. While this would impact the overall size of the ROIs used in the study, the remaining analyses are methodologically sound.

We have updated the text to detail the t=1.97 (i.e., p = .05) threshold we applied before interpretation of the resultant TFCE images to the section: *Experimental Design & Statistical Analysis.* This threshold value can also be verified in the analytics code that is referenced on GitHub from the section *Data Availability* within the requisite toolbox functions: https://github.com/kamentsvetanov/CommonalityAnalysis/blob/main/code/ca_vba_tfce_threshold.m#L24 and https://github.com/kamentsvetanov/CommonalityAnalysis/blob/main/code/external/ca_matlab_tfce_transform.m

Page-30

“For whole-brain voxelwise analyses, clusters were estimated using threshold-free cluster enhancement (TFCE; Smith & Nichols 2009) with 2000 permutations and the resulting images were thresholded at a t-statistic of 1.97 before interpretation.”

Point 3: The authors should consider moving the ROI section to results. The way the manuscript currently reads, the ROIs seem to be derived a priori as opposed to being derived from activation maps in the current study.

After consideration of this point, we have decided to leave the methodological details regarding the definition of ROIs in the methods, to maintain the focus of the Results section. However, we have improved signposting in the results section to highlight that the ROIs were derived from the overlapped activation maps.

Page-8

“Crucially, two areas of the brain showed spatially-overlapping positive effects of age and performance, which is suggestive of an age-related compensatory response (Figure 2A yellow intersection). These were in bilateral cuneal cortex (Figure 2B magenta) and bilateral frontal cortex (Figure 2B brown), the latter incorporating parts of the middle frontal gyri and anterior cingulate. Therefore, based on traditional univariate analyses, these are two candidate regions for age-related functional compensation (Cabeza et al. 2013; 2018). Accordingly, we defined regions of interest within these two regions using the overlap activation maps (see section: ROIs) to be used for subsequent univariate and multivariate analysis.”

Point 4: The manuscript can be strengthened by explaining why the authors chose a greedy search algorithm over a dynamic Bayesian model.

The text is updated to refer to appropriateness of the computationally efficient greedy search implementation, due to the size of the fMRI cohort dataset.

Page-28

“The pattern weights specifying the mapping of data features to the target variable are optimized with a greedy search algorithm using a standard variational scheme (Friston et al., 2007) which was particularly appropriate given the large dataset.”

**Reviewer #2:**
Point 1: However, it might have been nice to see an analysis of a more crystallised intelligence task included too, as a contrast since this is an area that does not demonstrate such a decline (and perhaps continues to improve over aging).

We (Samu et al., 2017) have previously investigated, but failed to find, univariate evidence for functional compensation in this cohort’s performance on a sentence comprehension task that is more closely aligned to a measure of crystallised intelligence. Based on the additional previous studies where we have applied these types of univariate and multivariate criteria of functional compensation (Morcom & Henson, 2018; Knights et al., 2021), we have consistently observed that the uni-/multivariate effects are in the same direction. Therefore, we would not initially expect a different conclusion here, where the univariate and multivariate effects suggest different outcomes. Notably, the univariate analysis approach in Samu et al. (2017) did differ from focusing on the age x behaviour interaction term here, so it could still be worth future investigation, but it does seem less likely that evidence of compensation would be observed than for fluid intelligence. However, as the Reviewer suggests, such a task may make another good contrast to show evidence against the existence of functional compensation (as in Morcom & Henson, 2018; Knights et al., 2021).

Point 2: Figure 1B: Consider adding coefficients describing relationships to plots.

Annotations of the coefficients have been added to Figure 1B:

Point 3: Figure 2C. The scale of the axis for RSFA-Scales cuneal cortex ROI activations should be the same as the other 3 plots.

Figure axes are updated such that ROIs are on matching scales, according to whether data were RSFA-scaled or not.

Point 4: Figure 2C. Adding in the age ranges for each of the three groups following the tertile split may be informative to the reader.

The age group tertile definition used for Figure 2C visualisations is now added to the Figure description.

Page-10

“Figure 2. Univariate analysis. (A) Whole-brain effects of age and performance. Age (green) and performance (red) positively predicted unique aspects of increased task activation, with their spatial overlap (yellow) being overlaid on a template MNI brain, using p < 0.05 TFCE. (B) Intersection ROIs. A bilateral cuneal (magenta) and frontal cortex (brown) ROI were defined from voxels that showed a positive and unique effect of both age and performance (yellow map in Figure 2A). (C) ROI Activation. Activation (raw = left; RSFA-scaled = right) is plotted against behavioural performance based on a tertile split between three age groups (19-44, 45-63 & 64-87 years).”

**Reviewer #3:**
Point 1: [Public Review] (1) I don't quite follow the argumentation that compensatory recruitment would need to show via non-redundant information carried by any given non-MDN region (cf. p14). Wouldn't the fact that a non-MDN region carries task-related information be sufficient to infer that it is involved in the task and, if activated increasingly with increasing age, that its stronger recruitment reflects compensation, rather than inefficiency or dedifferentiation? Put differently, wouldn't "more of the same" in an additional region suffice to qualify as compensation, as compared to the "additional information in an additional region" requirement set by the authors? As a consequence, in my honest opinion, showing that decoding task difficulty from non-MDN ROIs works better with higher age would already count as evidence for compensation, rather than asking for age-related increases in decoding boosts obtained from adding such ROIs. It would be interesting to see whether the arguably redundant frontal ROI would satisfy this less demanding criterion. At any rate, it seems useful to show whether the difference in log evidence for the real vs. shuffled models is also related to age.

We agree with the logic for conducting a weaker assessment of functional compensation whereby a brain region does not necessarily have to provide a unique contribution beyond that of the ordinarily activated task-relevant network. However, although non-unique recruitment is predicted by a compensation theory, it can also be explained by a nonspecific mechanism that recruits multiple regions in tandem. In contrast, unique additional recruitment is compatible with compensation but not with nonspecific recruitment. In this article, and those prior (Morcom & Henson, 2018; Knights et al. 2021), we have also deliberately avoided using the specific kind of analysis proposed (i.e., testing for an effect of age on differential log evidence) because these would involve applying statistical tests directly to the log evidence, a variable that is already a statistical test output.

Nevertheless, temporarily putting these caveats aside, we did run the suggested test. Results from multiple regression showed that using log evidence from frontal cortex models still did not meet this less demanding criterion for functional compensation as there was an effect of age in the opposite direction to that expected by functional compensation: there was a significant negative effect of age (*t*(218) = -7.95, *p* = < .001) indicating that as age increased, the difference in log evidence decreased. This effect is visualised below for transparency, but we preferred not to add this information to the article because we do not wish to encourage using this kind of analysis for the reason mentioned above. Thus, although our main multivariate test of interest is stringent, the additional step of mapping log evidence back to the boost-likelihood categories (e.g., boost vs. no difference to model performance) lends itself to the more appropriate logistic regression statistical approach.

**Author response image 1. sa4fig1:** Negative effect of age on MVB log evidence model outcomes for frontal cortex.

A different approach that could be taken to assess a more lenient definition of functional compensation would be to analyse the effects of age on the spread of multivariate responses predicting task difficulty (i.e., standard deviation of fitted MVB voxel weights; also see Morcom & Henson, 2018; Knights et al., 2021) specifically from models that only include the candidate ‘compensation’ ROIs.

Accordingly, these analyses and their discussion have been added to the article. To summarise, these analyses showed that (1) the frontal cortex still did not show evidence of functional compensation (i.e., a negative effect of age like in Morcom & Henson, 2018) and (2) no effect of age on the cuneal ROI, implying that the original model comparison approach (i.e., Figure 2C in the manuscript now) can provide more sensitivity for detecting evidence of functional compensation (perhaps because of the importance of including task-relevant network responses when building decoding models).

Page-15

“As a final analysis, we also tested a more lenient definition of functional compensation, whereby the multivariate contribution from the “compensation ROI” does not necessarily need to be above and beyond that of the task-relevant network (Morcom & Henson, 2018; Knights et al., 2021). To do this, we again assessed whether age was associated with an increase in the spread (standard deviation) of the weights over voxels, for smaller models containing only the cuneal or frontal ROI. This tested whether increased age led to more voxels carrying substantial information about task difficulty, a pattern predicted by functional compensation (but also consistent with non-specific additional recruitment). In this case, the results of this test did not support functional compensation, as there was no effect detected for the cuneal cortex and even a negative effect of age for the frontal cortex where the spread of the information across voxels was lower for older age (Figure 3C; Table 2).”

Page-21

“The age- and performance-related activation in our frontal region satisfied the traditional univariate criteria for functional compensation, but our multivariate (MVB) model comparison analysis showed that additional multivariate information beyond that in the MDN was absent in this region, which is inconsistent with the strongest definition of compensation. In fact, the results from the spread analysis showed that as age increased, this frontal area processed less, rather than more, multivariate information about the cognitive outcome (Figure 3C) as previously observed in two (memory) tasks for a comparable ROI within the same Cam-CAN cohort (Morcom & Henson, 2018).”

Page-24

“This said, univariate criteria for functional compensation will continue to play a role in hypothesis testing. For instance, the over-additive interaction observed in the cuneal cortex - where the increase in activity with better performance is more pronounced in older adults - offers stronger evidence of compensation compared to the simple additive effect of age and performance observed in the frontal cortex (Figure 2C). So far, the two studies that have combined these rigorous univariate, behavioral and multivariate approaches to assess functional compensation (i.e., Knights et al., 2021; the present study) have generally found converging evidence regardless of the method used. However, it is important to note that the MVB approach uniquely shifts the focus from individual differences to the specific task-related information that compensatory neural activations are assumed to carry and provides a specific test of region- (or network-) unique information. With further studies, it may also be that multivariate approaches prove more sensitive for detecting compensation effects than when using mean responses over voxels (e.g., Friston et al., 1995) particularly since over-additive effects are challenging to observe because compensatory effects are typically ‘partial’ and do not fully restore function (for review see Scheller et al., 2014; Morcom & Johnson, 2015). Within the multivariate analysis options themselves, it is also interesting to highlight that the stringent MVB boost likelihood analysis could detect functional compensation unlike the more lenient analysis focusing on the spread of MVB voxel weights. This suggests the importance of including task-relevant network responses when building decoding models to assess compensation.”

Page-32

“Alongside the MVB boost analysis, we also included an additional measure using the spread (standard deviation) of voxel classification weights (Morcom & Henson, 2018). This measure indexes the absolute amplitude of voxel contributions to the task, reflecting the degree to which multiple voxels carry substantial task-related information. When related to age this can serve as a multivariate index of information distribution, unlike univariate analyses. However, it is worth highlighting that even if an ROI shows an effect of age on this spread measure, such an effect could instead be explained by a non-specific mechanism that represents the same information in tandem across multiple regions (rather than reflecting compensation) as seen previously (Knights et al., 2021; also see Morcom & Johnson, 2015). Thus, it is the MVB boost analysis that is the most compelling assessment of functional compensation because it can directly detect novel information representation.”

Point 2: [Public Review] (2) Relatedly, does the observed boost in decoding by adding the cuneal ROI (in older adults) really reflect "additional, non-redundant" information carried by this ROI? Or could it be that this boost is just a statistical phenomenon that is obtained because the cuneus just happens to show a more clear-cut, less noisy difference in hard vs. easy task activation patterns than does the MDN (which itself may suffer from increased neural inefficiency in older age), and thus the cuneaus improves decoding performance without containing additional (novel) pieces of information (but just more reliable ones)? If so, the compensation account could still be maintained by reference to the less demanding rationale for what constitutes compensation laid out above.

We agree that this is a possibility and have added this as an additional explanation to the Discussion. We have also discussed why we think it is a less likely possibility, but do concede that it cannot be ruled out currently.

Page-20

“Another possibility is that the age-related increases in fMRI activations (for hard versus easy) in one or both of our ROIs do not reflect greater fMRI signal for hard problems in older than younger people, but rather lower fMRI signal for easy problems in the older. Without a third baseline condition, we cannot distinguish these two possibilities in our data. However, a reduced “baseline” level of fMRI signal (e.g., for easy problems) in older people is consistent with other studies showing an age-related decline in baseline perfusion levels, coupled with preserved capacity of cerebrovascular reactivity to meet metabolic demands of neuronal activity at higher cognitive load (Calautti et al., 2001; Jennings et al., 2005). Though age-related decline in baseline perfusion occurs in the cuneal cortex (Tsvetanov et al., 2021), the brain regions showing modulation of behaviourally-relevant Cattell fMRI activity by perfusion levels did not include the cuneal cortex (Wu et al., 2023). This suggests that the compensatory effects in the cuneus are unlikely to be explained by age-related hypo-perfusion, consistent with the minimal effect here of adjusting for RSFA (Figure 2C).

One final possibility is whether the observed boost in decoding from adding the cuneal ROI simply reflects less noisy task-related information (i.e., a better signal-to-noise ratio (SNR)) than the MDN and, consequently, the boosted decoding is the result of more resilient patterns of information (rather than the representation of additional information) based on a steeper age-related decline of SNR in the MDN. Overall then, as none of the explanations above agree with all aspects of the results, to functionally explain the role of the cuneal cortex in this task would require further investigation.”

Point 3: [Public Review] (3) On page 21, the authors state that "...traditional univariate criteria alone are not sufficient for identifying functional compensation." To me, this conclusion is quite bold as I'd think that this depends on the unvariate criterion used. For instance, it could be argued that compensation should be more clearly indicated by an over additive interaction as observed for the relationship of cuneal activity with age and performance (i.e., the activity increase with better performance becomes stronger with age), rather than by an additive effect of age and performance as observed for the prefrontal ROI (see Fig. 2C). In any case, I'd appreciate it if the authors discussed this issue and the relationship between univariate and multivariate results in more detail (e.g. how many differences in sensitivity between the two approaches have contributed), in particular since the sophisticated multivariate approach used here is not widely established in the field yet.

We have now considered this point further in a section of the Discussion (which is merged with points 1 & 2 above) about the relevance and distinction of univariate / multivariate criteria for functional compensation. As described in text below, whilst we agree that univariate / behavioural approaches have a role in testing functional compensation, we still view the MVB boost analysis to be a particularly compelling approach for assessing this theory.

Page-22

“This said, univariate criteria for functional compensation will continue to play a role in hypothesis testing. For instance, the over-additive interaction observed in the cuneal cortex - where the increase in activity with better performance is more pronounced in older adults - offers evidence of compensation compared to the simple additive effect of age and performance observed in the frontal cortex (Figure 2C). However, the conclusions that can be drawn from age-related differences in cross-sectional associations of brain and behaviour are limited, mainly because individual performance differences are largely lifespan-stable (see Lindenberger et al., 2011; Morcom & Johnson, 2015). So far, the two studies that have combined these univariate-behavioral and multivariate approaches to assess functional compensation (i.e., Knights et al., 2021; the present study) have generally found converging evidence regardless of the method used. However, it is important to note that the MVB approach uniquely shifts the focus from individual differences to the specific task-related information that compensatory neural activations are assumed to carry. With further studies, it may also be that multivariate approaches prove more sensitive for detecting compensation effects than when using mean responses over voxels (e.g., Friston et al., 1995) particularly since over-additive effects are challenging to observe because compensatory effects are typically ‘partial’ and do not fully restore function. Within the multivariate analysis options themselves, it is also interesting to highlight that the stringent MVB boost likelihood analysis could detect functional compensation unlike the more lenient analysis focusing on the spread of MVB voxel weights. This suggests the importance of including task-relevant network responses when building decoding models to asses compensation.”

Point 4: [Public Review] (4) As to the exclusion of poorly performing participants (see p24): If only based on the absolute number of errors, wouldn't you miss those who worked (overly) slowly but made few errors (possibly because of adjusting their speed-accuracy tradeoff)? Wouldn't it be reasonable to define a criterion based on the same performance measure (correct - incorrect) as used in the main behavioural analyses?

This is a good point, though if we were to exclude participants using a chance level exclusion rate based on the formulae used for measuring behavioural performance, this removes identical subjects to those originally excluded. Based on this, the text has been updated to reflect this more parsimonious approach for defining exclusion criteria.

Page-25

“In a block design, participants completed eight 30-second blocks which contained a series of puzzles from one of two difficulty levels (i.e., four hard and four easy blocks completed in an alternating block order; Figure 1A). The fixed block time allowed participants to attempt as many trials as possible. Therefore, to balance speed and accuracy, behavioural performance was measured by subtracting the number of incorrect from correct trials and averaging over the hard and easy blocks independently (i.e., ((hard correct - hard incorrect) + (easy correct - easy incorrect))/2; Samu et al., 2017). For assessing reliability and validity, behavioural performance (total number of puzzles correct) was also collected from the same participants during a full version of the Cattell task (Scale 2 Form A) administered outside the scanner at Stage 2 of the Cam-CAN study (Shafto et al., 2014). Both the in- and out-of-scanner measures were z-scored. We excluded participants (N = 28; 17 females) who performed at chance level ((correct + incorrect) / incorrect < 0.5) on the fMRI task, leading to the same subset as reported in Samu et al. (2017).”

Point 5: [Public Review] (5) Did the authors consider testing for negative relationships between performance and brain activity, given that there is some literature arguing that neural efficiency (i.e. less activation) is the hallmark of high intelligence (i.e. high performance levels in the Cattell task)? If that were true, at least for some regions, the set of ROIs putatively carrying task-related information could be expanded beyond that examined here. If no such regions were found, it would provide some evidence bearing on the neural efficiency hypothesis.

No, we did not test for negative relationships between performance and brain activity in this study. However, In Wu et al. (2023) we did specifically test for this and neither of the relevant results reported in section 3.3.1 (i.e., unique relationship between activity and performance) nor section 3.3.2 (i.e., age-related relationship between activity and performance) showed the queried direction of effects. Note that the negative effect in section 3.3.2 (Age U Performance) is a more unique suppression effect representing a positive relationship between performance and activity where this becomes stronger as age is added to the model.

Point 6: [Recommendations for the authors] (1) Page 26: It is not quite clear how the authors made sure their age and performance covariates functioned as independent regressors in the univariate group-level GLM, given the correlation between age and performance (i.e. shared variance).

We included age and performance as covariates (of the age x performance effect of interest) by simply including these as independent regressors in the group-level GLM design matrix in addition to the interaction term (i.e., *activity ~ age*performance + covariates* equivalent to *activity ~ age:performance + age + performance + covariates;* Wilkinson & Roger 1973 notation), allowing us to examine the unique variance explained by each predictor (Table 1 and Table 2) and to control for their shared variance.

We should note that while the GLM approach we used accounts for unique and shared effects, it does not explicitly report shared effects in its standard output. To directly examine shared variance, one would need to employ commonality analysis. For reference, results from a commonality analysis on this task have been previously reported in Wu et al. (2023).

Prompted by this point, we have made some further minor improvements to help ensure our methodological steps are reproducible, as highlighted below.

Page-30

“Continuous age and behavioural performance variables were standardised and treated as linear predictors in multiple regression throughout the behavioural (Figure 1B), wholebrain voxelwise (Figure 1C/2A), univariate (Table 1; Figure 1B/2B) and MVB (Table 2; Figure 3) analyses. Throughout, sex was included as a covariate. The models, including interaction terms, can be described, according to Wilkinson & Roger’s (1973) notation, as *activity ~ age * performance + covariates* (which is equivalent to *activity ~ age:performance + age + performance + covariates*), allowing us to examine the unique variance explained by each predictor (Table 1) and to control for their shared variance. For whole-brain voxelwise analyses, clusters were estimated using threshold-free cluster enhancement (TFCE; Smith & Nichols 2009) with 2000 permutations and the resulting images were thresholded at a t-statistic of 1.97 before interpretation. Bonferroni correction was applied to a standard alpha = 0.05 based on the two ROIs (cuneal and frontal) that were examined. For Bayes Factors, interpretation criteria norms were drawn from Jarosz & Wiley (2014)”.

Point 7: [Recommendations for the authors] (2) Figure 3: I suggest changing the subheading in panel B to "Joint vs. MDN-only Model," in line with the wording in the main text.

The subheading of Figure 3B is updated as suggested to `Joint vs. MDN-only Model`.

Point 8: [Recommendations for the authors] (3) In Figures 1C and 2A, MNI z coordinates should be added to the section views. The appreciation of Figure 2B could be enhanced by adding some rendering with a saggital (medial and/or lateral) view.

The slice mosaics in Figure 1C and 2A are now updated with each slice’s MNI Z coordinates and mentioned in the figure descriptions.

Point 9: [Recommendations for the authors] (4) Page 7 (l. 135): What exactly is meant by "lateral occipital temporal cortex"?

The text is updated to specify the anatomical landmarks that were used for guidance when referring to activation within the lateral occipital temporal cortex, based on ROI criteria definitions used in Knights, Mansfield et al. (2021):

Page-7 Line-135:

“Additional activation was observed bilaterally in the inferior/ventral and lateral occipital temporal cortex (i.e., a cluster around the lateral occipital sulcus that extended anteriorly beyond the anterior occipital sulcus), likely due to the visual nature of the task.”

Point 10: [Recommendations for the authors] (5) On p18ff. (ll. 259-318) the authors discuss in quite some detail how the age-related decoding boost seen with the cuneus ROI can be functionally explained, but it seems like none of the explanations agrees with all aspects of the results. While this is not a major problem for the paper, it may be advisable if this part of the discussion ends with a clearer statement that this issue is not fully solved yet and provides material for future research.

A more direct sentence has been added to make it clear that future investigation will be needed to explain the role of the cuneal cortex here.

Page-20 Line-322:

“Another possibility is that the age-related increases in fMRI activations (for hard versus easy) in one or both of our ROIs do not reflect greater fMRI signal for hard problems in older than younger people, but rather lower fMRI signal for easy problems in the older. Without a third baseline condition, we cannot distinguish these two possibilities in our data. However, a reduced “baseline” level of fMRI signal (e.g., for easy problems) in older people is consistent with other studies showing an age-related decline in baseline perfusion levels, coupled with preserved capacity of cerebrovascular reactivity to meet metabolic demands of neuronal activity at higher cognitive load (Calautti et al., 2001; Jennings et al., 2005). Though age-related decline in baseline perfusion occurs in the cuneal cortex (Tsvetanov et al., 2021), the brain regions showing modulation of behaviourally-relevant Cattell fMRI activity by perfusion levels did not include the cuneal cortex (Wu et al., 2021). This suggests that the compensatory effects in the cuneus are unlikely to be explained by age-related hypo-perfusion, consistent with the minimal effect here of adjusting for RSFA (Figure 2C). Overall then, as none of the explanations above agree with all aspects of the results, to functionally explain the role of the cuneal cortex in this task will require further investigation.”

Point 11: [Recommendations for the authors] (6) The threshold choice for Bayesian log evidence (> 3) should be motivated in some more detail, rather than just pointing to a book reference, as there is no established convention in the field, the choice may depend on the type of data and/or analysis, and a sizeable part of the readership may not be deeply familiar with the particular Bayesian approach used here.

Text is updated to further clarify our motivation for using the log evidence BF>3 criterion:

Page-29

“The outcome measure was the log evidence for each model (Morcom & Henson, 2018; Knights et al., 2021). To test whether activity from an ROI is compensatory, we used an ordinal boost measure (Morcom & Henson, 2018; Knights et al., 2021) to assess the contribution of that ROI for the decoding of task-relevant information (Figure 3B). Specifically, Bayesian model comparison assessed whether a model that contains activity patterns from a compensatory ROI and the MDN (i.e., a joint model) boosted the prediction of task-relevant information relative to a model containing the MDN only. The compensatory hypothesis predicts that the likelihood of a boost to model decoding will increase with older age. The dependent measure, for each participant, was a categorical recoding of the relative model evidence to indicate the outcome of the model comparison. The three possible outcomes were: a boost to model evidence for the joint vs. MDN-only model (difference in log evidence > 3), ambiguous evidence for the two models (difference in log evidence between -3 to 3), or a reduction in evidence for the joint vs. MDN-only model (difference in log evidence < -3).These values were selected because a log difference of three corresponds to a Bayes Factor of 20, which is generally considered strong evidence (Lee & Wagenmakers, 2014). Further, with uniform priors, this chosen criterion (Bayes Factor > 3) corresponds to a p-value of p<~.05 (since the natural logarithm of 20 equals three, as evidence for the alternative hypothesis).”

Point 12: [Recommendations for the authors] (7) Adding page numbers would be helpful.

Page numbers have been added to the manuscript file – apologies for this oversight.

References

Green, E., Bennett, H., Brayne, C., & Matthews, F. E. (2018). Exploring patterns of response across the lifespan: The Cambridge Centre for Ageing and Neuroscience (Cam-CAN) study. *BMC Public Health*, *18*, 1-7.

Knights, E., Mansfield, C., Tonin, D., Saada, J., Smith, F. W., & Rossit, S. (2021). Hand-selective visual regions represent how to grasp 3D tools: brain decoding during real actions. *Journal of Neuroscience*, *41*(24), 5263-5273.

Samu, D., Campbell, K. L., Tsvetanov, K. A., Shafto, M. A., & Tyler, L. K. (2017). Preserved cognitive functions with age are determined by domain-dependent shifts in network responsivity. *Nature communications,* 8(1), 14743.

Shafto, M. A., Tyler, L. K., Dixon, M., Taylor, J. R., Rowe, J. B., Cusack, R., ... & Cam-CAN. (2014). The Cambridge Centre for Ageing and Neuroscience (Cam-CAN) study protocol: a cross-sectional, lifespan, multidisciplinary examination of healthy cognitive ageing. *BMC neurology*, *14*, 1-25.

Wu, S., Tyler, L. K., Henson, R. N., Rowe, J. B., & Tsvetanov, K. A. (2023). Cerebral blood flow predicts multiple demand network activity and fluid intelligence across the adult lifespan. *Neurobiology of aging*, *121*, 1-14.